# Long-Form Video-Language Pre-Training with Multimodal Temporal Contrastive Learning

**Yuchong Sun**[1],[*] **Hongwei Xue**[2],[*] **Ruihua Song**[1],[†] **Bei Liu**[3],[†] **Huan Yang**[3], **Jianlong Fu**[3]

[1]Renmin University of China, Beijing, China,
[2]University of Science and Technology of China, Hefei, China,
[3]Microsoft Research, Beijing, China,
[1]{ycsun, rsong}@ruc.edu.cn, [2]gh051120@mail.ustc.edu.cn,
[3]{bei.liu, huayan, jianf}@microsoft.com

## Abstract

Large-scale video-language pre-training has shown significant improvement in video-language understanding tasks. Previous studies of video-language pre-training mainly focus on short-form videos (i.e., within 30 seconds) and sentences, leaving long-form video-language pre-training rarely explored. Directly learning representation from long-form videos and language may benefit many long-form video-language understanding tasks. However, it is challenging due to the difficulty of modeling long-range relationships and the heavy computational burden caused by more frames. In this paper, we introduce a **L**ong-**F**orm **VI**deo-**LA**nguage pre-training model (LF-VILA) and train it on a large-scale long-form video and paragraph dataset constructed from an existing public dataset. To effectively capture the rich temporal dynamics and to better align video and language in an efficient end-to-end manner, we introduce two novel designs in our LF-VILA model. We first propose a Multimodal Temporal Contrastive (MTC) loss to learn the temporal relation across different modalities by encouraging fine-grained alignment between long-form videos and paragraphs. Second, we propose a Hierarchical Temporal Window Attention (HTWA) mechanism to effectively capture long-range dependency while reducing computational cost in Transformer. We fine-tune the pre-trained LF-VILA model on seven downstream long-form video-language understanding tasks of paragraph-to-video retrieval and long-form video question-answering, and achieve new state-of-the-art performances. Specifically, our model achieves 16.1% relative improvement on ActivityNet paragraph-to-video retrieval task and 2.4% on How2QA task, respectively. We release our code, dataset, and pre-trained models at https://github.com/microsoft/XPretrain.

## 1 Introduction

In recent years, research on video understanding has attracted extensive attention due to the huge amount of videos available everywhere in our daily life. Previous research works on video understanding [14, 15, 42, 46, 60] mainly focus on short-form video (i.e., $< 30$ seconds) analysis and the semantics are limited to certain types (e.g., actions, scenes). However, there are so many long-form videos (i.e., $> 30$ seconds) [50] in real scenarios. Human annotated labels (e.g., actions) are difficult to cover the rich semantic and dynamic information contained in those videos. On the other hand, the video-language pre-training paradigm provides a way to learn cross-modal representation from

---

[*]This work was performed when Yuchong Sun and Hongwei Xue were visiting Microsoft Research as research interns.

[†]Ruihua Song and Bei Liu are the corresponding authors.

36th Conference on Neural Information Processing Systems (NeurIPS 2022).

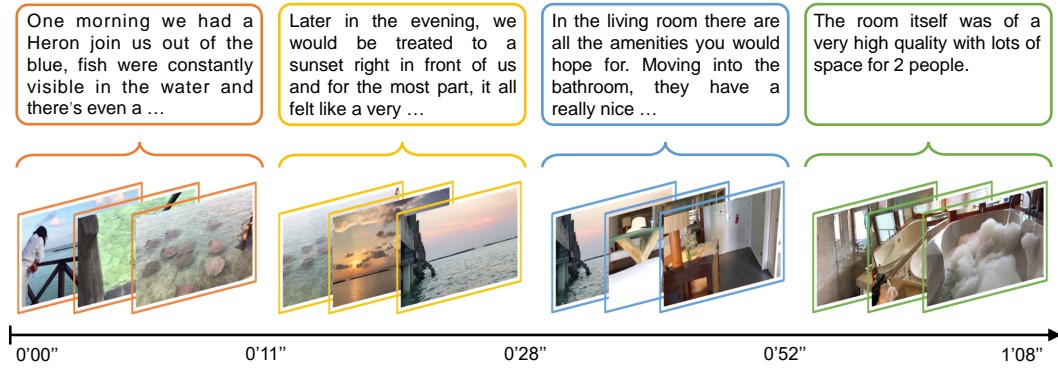

Figure 1: An example of long-form video-paragraph pair with several clips and sentences. It contains a complicated storyline and a rich temporal dynamic. Each sentence can only describe a short clip, and understanding the whole video needs the ability of long-range spatial-temporal reasoning.

video and language pairs and shows promising results on various high-level video understanding tasks joint with language [3, 27, 53, 58]. However, these studies mainly focus on short-form videos. In this paper, we explore directly exploiting long-form video and language pairs for pre-training to benefit a wide range of long-form video-language understanding tasks.

Although long-form video-language joint learning has been explored in downstream tasks [16, 27, 28, 30, 57, 59, 61], they either use pre-extracted video features which lead to the sub-optimal problem, or utilize image encoder to extract frame features that fail to model the long-range dependency in long-form videos. Recent works [3, 5, 33] have shown that a video Transformer [47] backbone helps to capture long-range dependency in an end-to-end fashion. An intuitive way for long-form video-language pre-training is to adopt a video Transformer based short-form video-language pre-training model [3, 53] with long-form data. However, there are two main challenges in such a design. First, long-form videos often contain more complicated storylines and richer temporal dynamics as shown in Fig. 1. Simply aligning video and paragraph using a vanilla contrastive loss like previous models [3, 53] will ignore the temporal relation between clips and sentences, thus hindering the quality of learned representation. Second, feeding more frames in a Transformer based video encoder will largely increase computational cost considering the self-attention operation.

To overcome the above challenges, we propose a **L**ong-**F**orm **VI**deo-**LA**nguage pre-training model (LF-VILA) with two novel designs. First, to better align long-form video-language pairs and learn the temporal relationship between visual and language modalities, we propose a Multimodal Temporal Contrastive (MTC) loss that learns temporal alignment between video clips and single sentences. MTC encourages similarity between two modalities to be consistent with their temporal relationship. In other words, the embedding distance between a video clip and a sentence closer in time should be smaller than its distance with sentences that are far in time. Combining with global alignment between video and paragraph, MTC ensures the model capture the temporal relation between video clips and single sentences and further helps to improve the quality of joint representation.

Second, to utilize the advantage of the Transformer for capturing long-range dependency while efficiently processing more frames for end-to-end training, we propose a Hierarchical Temporal Window Attention (HTWA) mechanism. As shown in Fig. 1, the frames sparsely sampled from a long-form video have large spatial and motion gaps, thus directly computing self-attention on all frames in all layers of the Transformer is inefficient and unnecessary. Instead, we only learn the attention between adjacent frames in the first few layers that focus more on details of spatial and temporal information. Then we gradually expand the window size in the following layers, where the high-level representation enables the model to better capture the relation between frames far apart. The computational cost is largely reduced with the proposed HTWA mechanism.

We conduct experiments and evaluate LF-VILA on seven downstream long-form video-language understanding tasks of paragraph-to-video retrieval and long-form video question-answering. We surpass the state-of-the-art models pre-trained on short videos by a large margin. Our results demonstrate the benefit of modeling long-range dependency for long-form videos. We also verify the effectiveness of our proposed MTC loss and HTWA mechanism through ablation studies.

Our contributions are summarized as follows: (1) We are the first to study end-to-end long-form video-language pre-training with large-scale video-paragraph data to benefit long-form video understanding. (2) We propose an MTC loss to capture the temporal relationship between clips and sentences while improving the joint representation of long-form video and language. (3) We design an HTWA mechanism for the video Transformer backbone, which can capture the long-range dependency in long-form videos effectively and efficiently. (4) We verify the effectiveness of our LF-VILA model on a wide range of downstream long-form video-language understanding tasks. Our model achieves state-of-the-art performance on four paragraph-to-video retrieval tasks and three long-form video question-answering tasks.

## 2 Related Work

### 2.1 Video Representation

Most previous video encoders utilize 3D-CNN based backbones [7, 46, 51]. These models show promising performance on short-form video understanding tasks, such as action classification and detection [7, 6, 17]. However, CNN has a limited receptive field and cannot effectively capture long-range dependency. Recent works have extended Vision Transformer [13] for video representation and demonstrated the benefit of long-range temporal learning [5, 33]. To reduce the computational cost, TimeSformer [5] introduces a factorized spacetime attention, while Video Swin-Transformer [32] restricts self-attention in a local 3D window. However, TimeSformer [5] is still computationally expensive when the number of input frames becomes large. Video Swin-Transformer [33] adopts a fix-sized temporal window which is not suitable for videos with large duration. We propose hierarchical temporal window attention to effectively learn the long-range dependency in long-form videos while reducing the computational cost.

### 2.2 Long-form Video Understanding

Long-form video understanding is less explored in previous studies. Some works use long-term context for improving recognition performance [41, 49]. Typical long-form video understanding tasks contain shot or event boundary detection [4] and temporal action detection [6], but these tasks cannot reveal the ability of a high-level understanding of the model. Jointly understanding long-form videos with language is a way to discover the rich semantics contained in videos and many benchmarks are proposed recently, such as paragraph-to-video retrieval [1, 2, 23, 37] and long-form video question-answering [27, 28, 30, 57]. Previous works that explore these tasks mostly use pre-extracted features, which hinder the performance because of sub-optimal features [16, 27, 59, 61]. We study end-to-end long-form video-language pre-training and transfer to long-form video-language understanding tasks.

### 2.3 Video-Language Pre-training

Inspired by the success of image-language pre-training [10, 19, 20, 21, 22, 39, 54], video-language pre-training is also explored recently. However, these works mainly focus on short-form videos [3, 34, 36, 53]. Some works use 3D-CNN as a video backbone [34, 36]. To utilize the advancement of Transformer, some works use sparsely sampled frames to reduce the computation requirements [3, 53]. One key factor for learning good representation is using contrastive loss to align multi-modal features [3, 34, 36, 53, 58]. We further design a multimodal temporal contrastive loss to conduct fine-grained alignment between long-form videos and paragraphs. The power of the pre-training model is largely dependent on the amount of training data, some works built large-scale video-language datasets [36, 53, 58], and we build a long-from video-paragraph dataset based on HD-VILA-100M [53]. There are several works have explored long-form video-language pre-training, HERO [27] uses pre-extracted features, while MELORT [58] uses an image encoder to separately encode frames which ignores joint spatial-temporal representation. Different from them, we use a video Transformer backbone and end-to-end pre-training on large-scale long-form video-paragraph datasets.

## 3 Approach

In this section, we first show the overall architecture of the proposed Long-Form VIdeo-LAnguage pre-training model (LF-VILA) in Sec. 3.1. Then we explain our proposed Multimodal Temporal

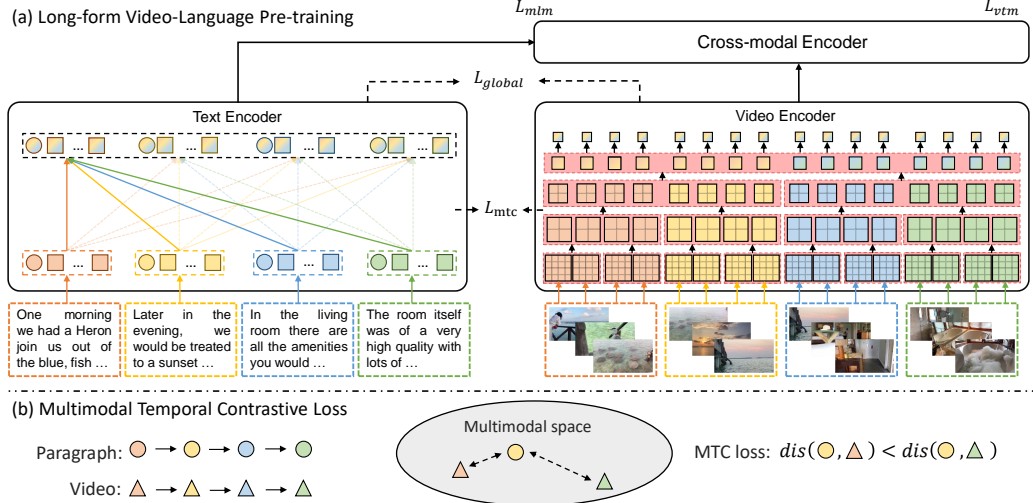

Figure 2: The framework of (a) Long-Form VIdeo-LAnguage pre-training model (LF-VILA) and illustration of (b) Multimodal Temporal Contrastive (MTC) learning. (a) LF-VILA consists of a text encoder, a video encoder, and a cross-modal encoder. In the text encoder, attention computing is first within each sentence then the whole paragraph. The pink boxes in the video encoder illustrate the proposed Hierarchical Temporal Window Attention (HTWA) mechanism. (b) the MTC loss aligns two sequences of representations (e.g., clip and sentence representations in our case), the distance of two element's representations is smaller when they are closer in time.

Contrastive (MTC) loss for learning cross-modal temporal relationship in Sec. 3.2, followed by the designed Hierarchical Temporal Window Attention (HTWA) mechanism for efficient video encoder in Sec. 3.3. Finally, we introduce the pre-training pipeline with target pre-training tasks in Sec. 3.4.

## 3.1 Model Architecture

As illustrated in Fig. 2, our proposed Long-Form VIdeo-LAnguage pre-training model (LF-VILA) consists of three parts: a video encoder $E_V$, a text encoder $E_T$ and a cross-modal encoder $E_C$. With a video and a paragraph as input, we first pass them to the video encoder $E_V$ and the text encoder $E_T$ for embedding learning, respectively. Then we concatenate visual and language embeddings as input to the cross-modal encoder $E_C$, where further cross-modal joint learning is conducted. The details of these encoders are as follows.

**Text Encoder.** The text encoder $E_T$ is based on Transformer network [47]. We divide it into two parts: sentence-level and paragraph-level encoding. In the first several layers, self-attention is conducted within word tokens from the same sentence, and sentence embedding can be learned individually. In higher layers, we add segment embedding to distinguish each sentence, and the attention computation is extended to all word tokens of the paragraph to output paragraph representation.

**Video Encoder.** Our video encoder is also stacked by Transformer layers. In particular, we design a Hierarchical Temporal Window Attention (HTWA) mechanism for efficient attention computation. Given a long-form video that has $M$ clips, we sample $N$ frames from each clip and divide each raw frame into $H \times W$ patches. Then the $M \times N \times H \times W$ patches are encoded by $E_V$. With our designed HTWA mechanism, the temporal window is gradually expanded, so that we can get hierarchical feature maps with different temporal receptive fields. In addition, video features with the same temporal window size as clip frame number $N$ in the middle layer can be utilized as the clip representation for fine-grained alignment with sentences.

**Cross-modal Encoder.** The cross-modal encoder $E_C$ consists of Transformer layers. Visual and language embeddings from the output of $E_V$ and $E_T$ are concatenated as the input to $E_C$. Self-attention is used to capture the joint relation between visual and language modalities. $E_C$ outputs the representation of [CLS] token and each textual and visual token.

## 3.2 Multimodal Temporal Contrastive Learning

Contrastive learning is widely used in previous multimodal pre-training works to align different modalities such as image-language and video-language. The goal of this loss is to pull the representation of matched pairs close to each other and push unmatched pairs away from each other. However, when aligning long-form videos and paragraphs, the vanilla contrastive loss neglects the temporal relation between clips and sentences, which is important for capturing the complex temporal dynamics in long-form videos. To better learn the temporal relationship between different modalities, we propose a Multimodal Temporal Contrastive (MTC) loss, which can be applied to align sequences in different modalities such as video and paragraph.

We assume that the distance of two elements' representations in different modalities should be consistent with their temporal distance. Specifically in our model, MTC loss encourages the video clip embedding to be more similar to its neighbor sentence embedding than sentences of long distance in the same paragraph. For example in Fig. 2, given two sequences of representation $v_i = \{v_i^1, v_i^2, ..., v_i^M\}$ and $t_i = \{t_i^1, t_i^2, ..., t_i^M\}$ from the $i$-th sample, we first sample an anchor set $\mathcal{A}$ of $k$ representations from $v_i$, then we sample a set of representations $\mathcal{K}$ from $t_i$. For each $v_i^p$ in $\mathcal{A}$, we treat $t_i^{q^+}$ as positive, where $|p - q^+| = \min(|p - q|), t_i^q \in \mathcal{K}$ [3]. We also randomly sample some representations from $t_j, j \neq i$ as $\mathcal{N}$ which is used to stable the training. Then the MTC loss is calculated by applying an InfoNCE loss:

$$\mathcal{L}_{mtc}(v_i, t_i) = -\frac{1}{k} \sum_{v_i^p \in \mathcal{A}} \log \frac{\exp\left(s(v_i^p, t_i^{q^+})\right)}{\sum_{t_i^q \in \mathcal{K}} \exp\left(s(v_i^p, t_i^q)\right) + \sum_{t_j^q \in \mathcal{N}} \exp\left(s(v_i^p, t_j^q)\right)}, \qquad (1)$$

where $s(f_1, f_2) = f_1^T \cdot f_2 / \tau$, $\tau$ is the temperature.

We obtain clip representations of $v_i$ and sentence representations $t_i$ from the output of the first part of the video encoder and text encoder, respectively. Then the MTC loss can be obtained by:

$$\mathcal{L}_{mtc}^{v2t} = -\frac{1}{B} \sum_{i=1}^{B} \mathcal{L}_{mtc}(v_i, t_i), \ \mathcal{L}_{mtc}^{t2v} = -\frac{1}{B} \sum_{i=1}^{B} \mathcal{L}_{mtc}(t_i, v_i), \qquad (2)$$

the overall MTC loss $\mathcal{L}_{mtc}$ is the average of $\mathcal{L}_{mtc}^{v2t}$ and $\mathcal{L}_{mtc}^{t2v}$.

## 3.3 Hierarchical Temporal Window Attention

Directly feeding all sampled frames of a long-form video to a vanilla Transformer network [47] for global self-attention learning will heavily increase the computational cost. The cost is quadratic with respect to the number of frames. One possible way is to apply a small fixed temporal window. However, such a design will neglect to learn the relationship between different clips of one video, which is essential for long-form video understanding. To capture long-range dependency in long-form videos efficiently and effectively, we propose a Hierarchical Temporal Window Attention (HTWA) mechanism by gradually increasing temporal window size in Transformer layers.

A temporal window is used to restrict attention computing between frames in the temporal dimension. Given a 3-D input tensor of $T' \times H' \times W'$ patches, where $T'$ is the number of time steps and $H' \times W'$ denotes the number of spatial patches. We divide the input tensor to $M_l$ windows along $T'$ dimension, where $l$ denotes the Transformer layer. Then multi-head attention is applied within each window, and the output is the combination of all attention windows as follows:

$$\begin{aligned} a_i &= MHA\left(z_i\right), i \in [1, 2, ..., M_l], \\ a &= Concat\left(a_1, a_2, ..., a_{M_l}\right), \end{aligned} \qquad (3)$$

where $z_i$ is the token embeddings belonging to $i$-th window, $MHA$ is multi-head attention, $a_i$ is the attention weights of $i$-th window, and $a$ is the attention weights with the shape of $T' \times H' \times W'$.

Compared with short-form videos, long-form videos have two distinguishing characteristics that we should consider for better understanding. First, there are large motion gaps (dynamics) between

---

[3]Since we consider $M = 4$ in this work, here $\mathcal{A}$ and $\mathcal{K}$ are randomly sampled. When we extend this loss for much longer sequences, we need to restrict the maximum distance between positive pairs.

frames that are sparsely sampled. Second, it is important to build long-range relationships to understand the whole video. Considering the above properties, we start to build connections between adjacent frames within small window sizes (e.g., 2) in the first few layers. Then the temporal window size is gradually increased to capture longer-range dependency as the semantics learned become high-level. Specifically, we use the temporal window size equal to the number of frames in a video in the last several layers, so that the full context can be attended to learn the global relationship.

### 3.4 Pre-training Pipeline

We adopt a two-stage pre-training as previous works did [53] since modal-independent design enables to learn powerful single-modality embedding for downstream tasks. In the first stage, the video encoder and text encoder are learned independently with a video-text alignment task. In the second stage, text embedding and video embedding are concatenated as input to the cross-model encoder for joint representation learning. We adopt Masked Language Modeling (MLM) and Video-Text Matching (VTM) which are widely used as pre-training tasks for learning cross-modal interaction.

**Video-Text Alignment.** Specifically, we first train the text-encoder and video-encoder using contrastive loss to align textual and visual representations. In addition to the proposed multimodal temporal contrastive loss, we also adopt a standard contrastive loss on the global representation of long-form video and paragraph. The global contrastive loss is calculated as:

$$\mathcal{L}_{global}^{v2t} = -\frac{1}{B} \sum_{i=1}^{B} \log \frac{\exp\left(s(V_i, T_i)\right)}{\sum_{j=1}^{B} \exp\left(s(V_i, T_j)\right)}, \quad \mathcal{L}_{global}^{t2v} = -\frac{1}{B} \sum_{i=1}^{B} \log \frac{\exp\left(s(T_i, V_i)\right)}{\sum_{j=1}^{B} \exp\left(s(T_i, V_j)\right)}, \quad (4)$$

where $V_i$ and $T_i$ are the representation of $i$-th video and paragraph in the batch, respectively. The global alignment loss $\mathcal{L}_{global}$ is the average of $\mathcal{L}_{global}^{v2t}$ and $\mathcal{L}_{global}^{t2v}$.

The combination of MTC loss and global contrastive loss is used as the pre-training objective for the first stage:

$$\mathcal{L}_{stage1} = \mathcal{L}_{global} + \lambda_1 \mathcal{L}_{mtc}, \quad (5)$$

where $\lambda_1$ denotes the weight of MTC loss compared with global contrastive loss.

**Masked Language Modeling.** We follow the previous vision-language pre-training works [20, 26, 54] to mask word tokens and predict the ground-truth labels from the output of the cross-modal encoder, which integrates the context of other textual tokens and visual tokens:

$$\mathcal{L}_{mlm} = -\mathbb{E}_{(\mathcal{W},\mathcal{V})} \log p\left(w_i \mid \mathcal{W}_{\backslash i}, \mathcal{V}\right), \quad (6)$$

where $\mathcal{W}$ denotes the word tokens, $\mathcal{V}$ denotes the visual tokens, and $w_i$ denotes the masked token. We adopt the same masking strategy and prediction method as BERT [12].

**Video-Text Matching.** To fuse the textual and visual information and generate a cross-modal representation, we use VTM as one pre-training task. VTM predicts whether the input paragraph and video are matched. We randomly replace the aligned video with a sampled negative with a probability of 0.5. We use a projection layer on the top of [CLS] embedding to predict two-class matching logits $y$, and then compute negative log likely-hood loss as the VTM loss:

$$\mathcal{L}_{vtm} = -\mathbb{E}_{(\mathcal{W},\mathcal{V})} \log p\left(y \mid \mathcal{W}, \mathcal{V}\right). \quad (7)$$

In the second stage, we freeze the video-encoder and text-encoder as [53] to accelerate training. The overall loss of stage two is the combination of MLM and VTM:

$$\mathcal{L}_{stage2} = \mathcal{L}_{mlm} + \lambda_2 \mathcal{L}_{vtm}, \quad (8)$$

where $\lambda_2$ is the weight of VTM in consideration of MLM.

## 4 Experiments

In this section, we first introduce the pre-training details and then show experiments of utilizing the pre-trained model on downstream paragraph-to-video retrieval and long-form video question-answering tasks to verify the effectiveness of our proposed LF-VILA. We also transfer LF-VILA for long-form video classification tasks to show the generalization power of the pre-trained model.

Table 1: Statistics of LF-VILA-8M and its comparison with existing video-language datasets.

| Dataset | Domain | #Video-Text Pairs | Avg. Len(sec) | Text Len | Duration(h) |
|---|---|---|---|---|---|
| DiDeMo [1] | Flickr | 10K | 28.0 | 29.2 | 87 |
| QuerYD [37] | open | 2K | 278.0 | 243.8 | 200 |
| ActivityNet Captions [23] | action | 20K | 180.0 | 48.3 | 849 |
| Condensed Movie [2] | movie | 34K | 132.0 | 18.0 | 1.3K |
| WebVid-2.5M [3] | open | 2.5M | 18.0 | 12.0 | 13K |
| HowTo100M [36] | instruction | 136M | 3.6 | 4.0 | 135K |
| HD-VILA-100M [53] | open | 103M | 13.4 | 32.5 | 372K |
| LF-VILA-8M | open | 8.5M | 100.2 | 307.9 | 236K |

Table 2: Results of paragraph-to-video retrieval on ActivityNet Captions dataset [23].

| Method | Pre-training Dataset | R@1 ↑ | R@5 ↑ | R@50 ↑ | MedR ↓ |
|---|---|---|---|---|---|
| HSE [59] | - | 20.5 | 49.3 | - | - |
| ClipBERT [25] | COCO [9], Visual Genome [24] | 21.3 | 49.0 | - | 6.0 |
| HD-VILA [53] | HD-VILA-100M [53] | 28.5 | 57.4 | 94.0 | 4.0 |
| Frozen [3] | CC3M, WebVid-2.5M [3] | 28.8 | 60.9 | - | 3.0 |
| Support Set [38] | HowTo100M [36] | 29.2 | 61.6 | 94.7 | 3.0 |
| TACo [56] | HowTo100M [36] | 30.4 | 61.2 | 93.4 | 3.0 |
| LF-VILA (Ours) | LF-VILA-8M | **35.3** | **65.4** | **95.0** | **3.0** |

## 4.1 Pre-training Details

**Pre-training Dataset.** To facilitate research on long-form video understanding, We build a large-scale long-form video-paragraph dataset based on HD-VILA-100M [53], which is an existing large-scale video-language dataset with diverse categories. It contains 100 million clip-sentence pairs derived from 3.3 million YouTube videos. We keep continuous clips with at least 4 clips and construct a dataset with 8.5 million long-from videos and corresponding transcripts, namely LF-VILA-8M. Table 1 shows the statistics of LF-VILA-8M and its comparison with existing video-language datasets. It covers ~60% of clips of the HD-VILA-100M dataset. The average duration of each video is 100.2 seconds and the average number of words in each paragraph is 307.9. We provide additional statistics and examples in the supplementary material.

**Implementation Details.** During pre-training, our model samples 4 consecutive clip-sentence pairs as input. We uniformly sample 8 frames from each clip and resize the frames to $192 \times 320$. we use the WordPiece tokenizer like BERT to split each sentence into tokens with a max length of 50. For the video encoder, we use Swin-Transformer [32] as the backbone and integrate our proposed HTWA for frame sequence. Temporal window sizes are set to five stages: 2, 4, 8, 16, and 32, respectively. We use $8 \times 8$ patches and use a fixed spatial window of $3 \times 5$, the output feature is down-sampled by 64 times to $3 \times 5$. We adopt a 12-layer Transformer network for the text encoder, with 8 layers for the first part and 4 layers for the second part. We also use a 12-layer Transformer network for the cross-modal encoder. The weight of the video encoder is initialized with Swin-Transformer pre-trained on ImageNet-21K. We use the first 12 layers of BERT-Large to initialize the weight of the text encoder, and the last 12 layers to initialize the weight of the cross-modal encoder. We provide more detailed model specifications in the supplementary material.

We use an AdamW optimizer with a learning rate of 5e-5 and warm up the learning rate for 1 epoch, followed by a linear decay, we use a weight decay of 0.05. We train our model with 32 NVIDIA Tesla V100 GPUs. For stage one, we use a batch size of 512 and train for 6 epochs. For stage two, we use a batch size of 1,536 and train for another 6 epochs. We use the model from stage one for retrieval tasks since two-stream architecture is efficient for retrieval and is widely used in previous works [39, 3, 53]. Pre-trained model from stage two is applied for video QA tasks. We excluded videos that overlap with downstream tasks from the training dataset using YouTube IDs.

Table 3: Results of paragraph-to-video retrieval on two datasets. * denotes results by our re-implementation.

(a) Result on DiDeMo dataset [1].

| Method | R@1↑ | R@5↑ | R@10↑ |
|---|---|---|---|
| FSE [59] | 13.9 | 36.0 | - |
| ClipBERT [25] | 20.4 | 48.0 | 69.0 |
| HD-VILA [53] | 28.8 | 57.4 | 69.1 |
| Frozen [3] | 31.0 | 59.8 | 72.4 |
| LF-VILA (Ours) | **35.0** | **64.5** | **75.8** |

(b) Result on QuerYD dataset [37].

| Method | R@1↑ | R@5↑ | R@10↑ |
|---|---|---|---|
| MOEE [35] | 11.6 | 30.2 | 43.2 |
| CE [31] | 13.9 | 37.6 | 78.3 |
| TeachText [11] | 14.4 | 37.7 | 50.9 |
| Frozen* [3] | 53.8 | 75.7 | 82.7 |
| LF-VILA (Ours) | **69.7** | **85.7** | **90.3** |

Table 4: Results of paragraph-to-video retrieval on Condensed Movie dataset [2] from official leaderboard at https://competitions.codalab.org/competitions/34124.

| Method | Geometric Mean ↑ | R@1↑ | R@5↑ | R@10↑ |
|---|---|---|---|---|
| MoEE [35] | 5.88 | 1.94 | 7.84 | 13.38 |
| TeachText [11] | 23.15 | 12.08 | 27.40 | 37.45 |
| LF-VILA (Ours) | **26.40** | **13.56** | **32.47** | **41.79** |

## 4.2 Paragraph-to-Video Retrieval

We conduct retrieval tasks on four widely-used datasets for paragraph-to-video retrieval: **ActivityNet Captions** [23], **DiDeMo** [1] , **QuerYD** [37] and **Condensed Movie** [2]. Details of each dataset and implementation are in the supplementary material.

**Results.** Tab. 2~ 4 show the results of LF-VILA on four paragraph-to-video retrieval datasets. For the most widely used **ActivityNet Captions** [23] dataset, we surpass the SOTA model TACo [56] by **16.1%** on R@1. TACo is pre-trained on HowTo100M [36] using pre-extracted feature. This demonstrates the benefit of end-to-end training and utilization of long-form video datasets. Compared to HD-VILA [53] which is trained on 100M short-form video and sentence pairs, we use long-form videos with fewer data (~60% clips of HD-VILA-100M) and achieve **23.9%** improvement in terms of R@1. This shows the effectiveness of our model LF-VILA in learning better alignment for long-form video and language. Compared to Frozen [3] which is pre-trained on a human-annotated dataset, we achieve **22.6%** improvement on R@1 using relatively noisy but easy-to-get data. For **DiDeMo** [1], we also observe a significant improvement. In particular, we obtain **12.9%** improvement in terms of R@1 over the SOTA model Frozen [3]. On **QuerYD** [37] and **Condensed Movie** [2], we outperform the previous best models Frozen [3] and TeachText [11] with a relative **29.6%** and **12.3%** improvement, respectively. These two datasets are challenging due to their long videos. These results show the value of long-form video-language pre-training and our LF-VILA model can better understand the storyline and temporal relations in long videos.

## 4.3 Long-Form Video Question-Answering

We conduct video question-answering tasks on three widely-used datasets for long-form video understanding: **ActivityNet-QA** [57], **How2QA** [27] and **VIOLIN** [30]. Details of each dataset and implementation are in the supplementary material.

**Results.** Tab. 5 shows the results of three long-form video question-answering tasks. For **ActivityNet QA** [57], we outperform most previous works except MELORT [58]. Note that VQA-T [55] and MERLOT [58] are specifically designed for video QA. VQA-T [55] uses automatically generates 69M video question-answer triplets from narrated videos for training. MERLOT [58] utilizes 180M pairs of data and it needs excessive computational cost for training (~30K TPU hours), while we only need ~5K GPU hours. For **How2QA** [27] and **VIOLIN** [30], we achieve the new SOTA. This illustrates the reasoning capability of our model on long-form videos by better capturing the long-range dependency between video clips and paragraphs.

Table 5: Results of video question-answering tasks. We gray out some results for fair comparison because of the use of large-scale video QA triplets or the huge computational cost for pre-training.

(a) ActivityNet QA [57].

| Method | Acc↑ |
|---|---|
| MAR-VQA [61] | 34.6 |
| CoMVT [40] | 38.8 |
| VQA-T [55] | 38.9 |
| MERLOT [58] | 41.4 |
| LF-VILA (Ours) | **39.9** |

(b) How2QA [27].

| Method | Acc↑ |
|---|---|
| CLIP [39] by [28] | 69.3 |
| CLIP-SF [28] | 72.9 |
| ResNet-SF [28] | 74.3 |
| HERO [27] | 74.3 |
| LF-VILA (Ours) | **76.1** |

(c) VIOLIN [30].

| Method | Acc↑ |
|---|---|
| LXMERT [44] | 66.3 |
| Det-BERT [30] | 67.8 |
| GVE [8] | 68.4 |
| HERO [27] | 68.6 |
| LF-VILA (Ours) | **70.9** |

Table 6: Results of Procedural Activities Classification on the COIN [45] dataset. We gray out some results for fair comparison because of the huge computational cost for pre-training.

| Model | Pre-training Dataset | #Training Samples | Domain | Acc↑ |
|---|---|---|---|---|
| ClipBERT [25] | COCO [9], Visual Genome [24] | 5.6M | Open | 65.4 |
| MIL-NCE [34] | HowTo100M [36] | 100M | Instructional | 70.2 |
| VideoCLIP [52] | HowTo100M [36] | 100M | Instructional | 72.5 |
| SlowFast [14] | Kinetics [7] | 370K | Action | 71.6 |
| TimeSformer [5] | Kinetics [7] | 370K | Action | 83.5 |
| TimeSformer [5] | HowTo100M [36] | 100M | Instructional | 85.3 |
| TimeSformer [5] | HowTo100M [36], wikiHow [29] | 100M | Instructional | 88.9 |
| LF-VILA (Ours) | LF-VILA-8M | 8M | Open | **85.7** |

## 4.4 Transfering to Long-form Video Classification

To demonstrate the improvement of LF-VILA for long-form video representation and broaden the tasks and domains of the evaluation, we evaluate LF-VILA on **COIN** [45] and **LVU** [50]. Details of each dataset and implementation are in the supplementary material.

Tab. 6 shows the result of Procedural Activities Classification on **COIN** [45] dataset. Our model achieves strong performance on this task, although we use out-domain videos for pre-training and our computational cost is smaller than the SOTA method (~2.1K vs. ~7K GPU hours). We largely surpass other video-language pre-training models (e.g., ClipBERT [25], MIL-NCE [34] and VideoCLIP [52]). Tab. 7 shows the result of video classification on **LVU** [50] dataset. Our model surpasses the previous SOTA methods largely, especially on Scene and Relation classification. Especially, we outperform the video-language pre-training model VideoBERT [43] largely. The strong performances on these two benchmarks show the generalization power of our pre-trained video encoder.

## 4.5 Ablation Studies

To validate the effectiveness of LF-VILA with MTC loss and HTWA mechanism, we conduct ablation studies with a subset of data to save resources. We randomly sample 1M video-paragraph pairs from the whole LF-VILA-8M for pre-training.

**(1) Using more frames is better for long-form video-language understanding.** Using the same backbone, we compare our model with pre-training with one clip-sentence pair for each sample as in previous models [3, 3]. As shown in Tab. 8, when only using 8 frames, the performance is poor. After increasing the number of frames to 32, there is a large improvement. This also indicates that an efficient backbone to support more frames is essential for long-form video representation.

**(2) Pre-training on long-form video-language data further improves the performance significantly.** When we use 4 continuous clips for pre-training, there is also a significant performance gain as shown in Tab. 8. The gain is larger on QuerYD dataset which consists of longer videos than ActivityNet dataset (278s vs 180s on average).

**(3) MTC loss contributes to the performance.** As shown in Tab. 8, when combined with the MTC loss, the performance is further improved by 1.7% on ActivityNet dataset and 1.9% on QuerYD dataset in terms of R@1.

Table 7: Results of Content Understanding on the LVU [50] dataset.

| Model | Relation (Acc)↑ | Way of Speaking (Acc)↑ | Scene (Acc)↑ |
|---|---|---|---|
| R101-SlowFast+NL [14, 18, 48] | 52.4 | 35.8 | 54.7 |
| VideoBERT [43] | 52.8 | 37.9 | 54.9 |
| Object Transformer [50] | 53.1 | 39.4 | 56.9 |
| LF-VILA (Ours) | **61.5** | **41.3** | **68.0** |

Table 8: Analysis of the effectiveness of long-form pre-training.

| Pretain | | | ActivityNet [23] | | | QuerYD [37] | | |
|---|---|---|---|---|---|---|---|---|
| Loss | #clips | #frames in total | R@1↑ | R@5↑ | R@50↑ | R@1↑ | R@5↑ | R@10↑ |
| w/o Pre-training | - | - | 15.0 | 40.2 | 85.8 | 19.8 | 46.4 | 58.4 |
| $\mathcal{L}_{global}$ | 1 | 8 | 19.3 | 45.5 | 87.1 | 38.4 | 67.5 | 78.2 |
| | 4 | 8 | 20.3 | 47.0 | 87.7 | 41.6 | 73.5 | 82.1 |
| | 1 | 32 | 24.3 | 53.5 | 92.1 | 50.8 | 75.0 | 83.4 |
| | 4 | 32 | 26.1 | 56.7 | 92.7 | 55.4 | 79.3 | 85.5 |
| $\mathcal{L}_{global} + \mathcal{L}_{mtc}$ | 4 | 32 | **27.8** | **58.3** | **92.8** | **57.3** | **80.9** | **85.8** |

**(4) HTWA mechanism achieves a better computational cost-performance tradeoff.** In Tab. 9, we compare our methods with video backbone using fixed window long-range dependency. When we apply a small fixed window size (e.g.,4), the performance is relatively poor. This indicates the limitation of a small attention window for modeling long-range dependency. When we increase the window size to 32 to cover a whole video, there is almost no improvement in performance, while the computational cost and training time increases significantly.

Table 9: Analysis of temporal window (TW).

| Method | TW# | Time | ActivityNet [23] | | | QuerYD [37] | | |
|---|---|---|---|---|---|---|---|---|
| | | | R@1↑ | R@5↑ | R@50↑ | R@1↑ | R@5↑ | R@10↑ |
| Fixed | 4 | 0.91× | 25.1 | 54.9 | 91.8 | 52.8 | 76.3 | 84.5 |
| | 8 | 0.96× | 25.4 | 54.8 | 91.9 | 54.5 | 78.1 | 84.4 |
| | 16 | 1.10× | 26.0 | 56.1 | 91.5 | 53.7 | 78.6 | 84.3 |
| | 32 | 1.49× | 26.2 | 56.4 | 92.3 | 55.6 | 79.3 | 85.7 |
| HTWA | [2-32] | 1.00× | 26.1 | 56.7 | 92.7 | 55.4 | 79.3 | 85.5 |

## 5   Conclusion

In this paper, we study video-language pre-training on a large-scale long-form video-paragraph dataset. To better align long-from videos and paragraphs, we propose a Multimodal Temporal Contrastive (MTC) loss to capture the rich temporal relation between different modalities. In addition, we design a Hierarchical Temporal Window Attention (HTWA) mechanism to be applied with an image Transformer. Our proposed Long-Form VIdeo-LAnguage pre-training model (LF-VILA) combined with MTC and HTWA can learn effective multi-modal representation by capturing long-range dependency from long-form videos efficiently. Experiments on seven long-form video-language understanding tasks verify the effectiveness of our model.

## Acknowledgments and Disclosure of Funding

Funding in direct support of this work: Fundamental Research Funds for the Central Universities and the Research Funds of Renmin University of China (21XNLG28). Additional revenues related to this work: Internship at Microsoft Research Asia. This work is also partially sponsored by Kuaishou Research Collaboration Initiative.

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
