# Supplementary Material: Long-Form Video-Language Pre-Training with Multimodal Temporal Contrastive Learning

**Yuchong Sun[1], Hongwei Xue[2], Ruihua Song[1], Bei Liu[3], Huan Yang[3], Jianlong Fu[3]**
[1]Renmin University of China, Beijing, China,
[2]University of Science and Technology of China, Hefei, China,
[3]Microsoft Research, Beijing, China,
[1]{ycsun, rsong}@ruc.edu.cn, [2]gh051120@mail.ustc.edu.cn,
[3]{bei.liu, huayan, jianf}@microsoft.com

This supplementary material is organized as:

## A Model Architecture Details

### A.1 Text Encoder Specifications

Our text encoder $E_T$ is a 12-layer Transformer [20] encoder, with a hidden size of 1024 and attention heads of 16. The parameter is initialized using the first 12 layers of BERT-Large [6] from HuggingFace [3] transformers library implementation. $E_T$ is divided into two parts, with 8 layers for the first part and 4 layers for the second part. $E_T$ takes a paragraph that has 4 sentences as input. We first split each sentence into tokens beginning with a [CLS] token using WordPiece tokenizer with a max length of 50. We add position embeddings to each sentence separately, and then encode each sentence using the first part of $E_T$ where attention computation is restricted to tokens belonging to the same sentence. We obtain representation for each sentence on the top of the [CLS] token which is used for MTC loss. At the beginning of the second part, we add segment embeddings to distinguish each sentence. We average the 4 [CLS] embeddings as a global [CLS] token and append it at the beginning of the token sequence. Then the attention is computed among all tokens in the second part, where the number of tokens is $1 + 4 \times 50$. At the top of the global [CLS] token, we obtain paragraph representation used for global alignment.

### A.2 Video Encoder Specifications

We apply the HTWA mechanism to an image Transformer, we choose Swin Transformer-base [12] here. The original Swin Transformer base is divided into 4 stages, we divide stage3 into 2 parts applied with different temporal windows. Tab. 1 shows the specific configuration, where"downs"

---

[*]This work was performed when Yuchong Sun and Hongwei Xue were visiting Microsoft Research as research interns.

[†]Ruihua Song and Bei Liu are the corresponding authors.

[3]https://huggingface.co/

36th Conference on Neural Information Processing Systems (NeurIPS 2022).

Table 1: Video Encoder Configuration.

| | Swin Transformer-base | | | Video Encoder of LF-VILA | |
|---|---|---|---|---|---|
| stage | downsp (size) | layer | stage | downsp (size) | layer |
| stage1 | $4\times$ $(56\times56)$ | concat $4 \times 4$, 128-d, [win.sz. $7 \times 7$] $\times 2$ | stage1 | $8\times$ $(32\times24\times40)$ | concat $8 \times 8$, 128-d, [win.sz. $2 \times 3 \times 5$] $\times 2$ |
| stage2 | $8\times$ $(28\times28)$ | concat $2 \times 2$, 256-d, [win.sz.$7 \times 7$] $\times 2$ | stage2 | $16\times$ $(32\times12\times20)$ | concat $2 \times 2$, 256-d, [win.sz. $4 \times 3 \times 5$] $\times 2$ |
| stage3 | $16\times$ $(14\times14)$ | concat $2 \times 2$, 512-d, [win.sz.$7 \times 7$] $\times 18$ | stage3 | $32\times$ $(32\times6\times10)$ | concat $2 \times 2$, 512-d, [win.sz. $8 \times 3 \times 5$] $\times 14$ |
| | | | stage4 | $32\times$ $(32\times6\times10)$ | 512-d, [win.sz. $16 \times 3 \times 5$] $\times 4$ |
| stage4 | $32\times$ $(7\times7)$ | concat $2 \times 2$, 1024-d, [win.sz.$7 \times 7$] $\times 2$ | stage5 | $64\times$ $(32\times3\times5)$ | concat $2 \times 2$, 1024-d, [win.sz. $32 \times 3 \times 5$] $\times 2$ |

means the times of spatial feature map downsampling,"siz" means the size of the feature map, "concat $n \times n$" means merging $n \times n$ neighboring spatial patches,"n-d" means the dimension of output feature,"win.sz" means a window attention module, the original Swin Transformer uses a fixed spatial window size of $7 \times 7$, and we use an additional temporal window with gradually expanding size. We initialize the video backbone using Swin Transformer-base parameters. There are two main modifications: (1) the patch projection layer, we divide the frame to $8 \times 8$ patches, while Swin Transformer uses $4 \times 4$ patches, so we duplicate projection weight 4 times. (2) the relative position embedding parameter, we first interpolate the original parameter to fit our spatial window size, then we duplicate 2d relative position embedding along the temporal dimension as [13]. The temporal window size is 8 in stage3, which means the attention computation is within a clip. We downsample the feature map of the stage3 two times and then average the tokens from the same clip to obtain clip representation which is used for MTC loss. After stage5, we average all tokens to obtain video representation used for global alignment.

### A.3 Cross-modal Encoder Specifications

The cross-modal encoder is also a 12-layer Transformer encoder, with a hidden size of 1024 and attention heads of 16. The parameter is initialized using the last 12 layers of BERT-Large [6] from HuggingFace transformers library implementation. We use $2 \times 3$ maxpool with stride $1 \times 1$ to downsample video feature maps and obtain 6 tokens for each frame, the number of video tokens is $32 \times 6$. We concatenate all the textual tokens and video tokens. Finally the tokens of cross-modal encoder is $1 + 50 \times 4 + 32 \times 6$.

### A.4 Video Encoder Computational Cost

We did not adopt the same backbone as HD-VILA [23] or Frozen [3] on the new dataset because their video encoders cannot be fed with so many frames in long-form videos of LF-VILA-8M while using more frames is critical for long-form video-language understanding. In Tab. 2, we measure the computational cost of our video encoder compared with HD-VILA and Frozen for encoding 8 or 32 frames.

Table 2: Comparison of video encoder computational cost.

| Model | #Frames | Flops | Memory |
|---|---|---|---|
| Frozen [3] | 8 | 356 G | 4.8G |
| HD-VILA [23] | 8 | 516 G | 6.3G |
| Frozen [3] | 32 | 1424 G | 11.6G |
| HD-VILA [23] | 32 | 1750 G | 13.1G |
| LF-VILA (Ours) | 32 | **298G** | **5.2G** |

Table 3: More details of LF-VILA and other pre-training models. * means using distilled text encoder.

| Model | Frozen [3] | HD-VILA [23] | LF-VILA (Ours) |
|---|---|---|---|
| Pre-training Dataset | CC3M [17], COCO [4], WebVid2.5M [3] | HD-VILA-100M [23] | LF-VILA-8M |
| #Training Examples | 6.1M | 100M | 8M |
| Pre-training Cost (GPU Hours) | ~1.3K | ~65K | ~2.1K |
| #Param | 223M(181M*) | 310M | 277M |
| Input Resolution | 224×224 | 640×1024 (1 frame), 160×256 | 192×320 |

# B  Additional Pre-training Details

## B.1  Hyperparameters

For the temperature parameter $\tau$ in $\mathcal{L}_{global}$ and $\mathcal{L}_{mtc}$, we set it to 0.05 as [3, 23]. The loss weight $\lambda_1$ in Equation 5 is set to 1.0 and the loss weight $\lambda_2$ in Equation 8 is set to 10.0 empirically. For the hyperparameters in $\mathcal{L}_{mtc}$, we set the size of $\mathcal{A}$ and $\mathcal{K}$ to 2 and the size of $\mathcal{N}$ to 3 empirically.

## B.2  More Details of LF-VILA and other Pre-training Models.

We provide more details about the pre-training cost, pre-training data, parameters and input resolution of our model and other large-scale end-to-end pre-training models in Tab. 3. Compared to Frozen [3] which is pre-trained on a human-annotated dataset, we achieve much better performance using relatively noisy but easy-to-get data. Compared to HD-VILA [23], we greatly reduce the training cost with smaller model size. LF-VILA-8M is a subset of HD-VILA-100M [23] (~60% clips). Thus we achieve better performance with less pre-training data, fewer parameters and lower input resolution.

# C  Downstream Task Details

## C.1  Paragraph-to-Video Retrieval

**Datasets.** We conduct retrieval tasks on four paragraph-video retrieval datasets as they are widely used in previous video-language pre-training works. **(1) ActivityNet Captions** [7] consist of 20K videos collected from YouTube and 100K manually annotated sentences. We follow [8, 23, 25] to concatenate all sentences of a video and evaluate paragraph-to-video retrieval. We fine-tune LF-VILA on the training set with 10K videos and report the result on the val1 set with 4.9K videos. **(2) DiDeMo** [1] consists of 10K Flickr videos and 40K manually annotated sentences. We conduct paragraph-to-video retrieval on this dataset following [3, 8, 23]. We use a standard split to fine-tune LF-VILA on the training set and report the result on the test set. **(3) QuerYD** [16] contains videos sourced from YouTube. There are 1815 videos in the training split, 388 and 390 videos for validation and testing, respectively. The dataset has 31,441 high-quality descriptions. We follow [5, 16] to

evaluate paragraph-level video-retrieval [4]. **(4) Condensed Movie** [2] consists of 35K key scenes from over 3.7K movies, each key scene is accompanied by a high-level semantic description. We follow the challenge instruction, fine-tune LF-VILA on 30K videos, and report results on the test set with 1K videos from the leaderboard.

**Implementation Details.** We adopt the pre-training model from stage one for fine-tuning because the two-stream architecture is suitable for retrieval tasks and is widely-adopted [3, 23]. We only use $\mathcal{L}_{global}$ for fine-tuning to keep a fair comparison with previous works [3, 23]. We sample 32 frames from the videos and resize the frames to $192 \times 320$. We merge adjacent short sentences in a paragraph to make the number of sentences 4. We use an AdamW optimizer with an initial learning rate of 5e-6 followed by a multi-step learning rate decay. We fine-tune the model with 8 NVIDIA Tesla V100 GPUs and use a batch size of 128.

## C.2    Long-Form Video Question-Answering

**Datasets.** We conduct video question-answering tasks on three widely-used datasets for long-form video understanding. **(1) ActivityNet QA** [24] is an open-ended question-answering dataset with 25K questions from 5.8K videos. The average video length is 180 seconds. It is a benchmark to test long-term spatial-temporal reasoning. We use the official dataset split and report the result on the test set. **(2) How2QA** [9] is a multiple-choice video question-answering dataset. It consists of 44K QA pairs from 22K 60-second clips selected from HowTo100M [15] dataset, each question has one correct answer and 3 wrong answers. We use the split of [10] and report the result on the validation set. **(3) VIOLIN** [11] is a video-language inference task. It aims to predict whether a video entails a hypothesis or contradicts the hypothesis when given a premise video with aligned subtitles and a hypothesis sentence. It consists of 95.3K video-hypothesis pairs from 15.9K video clips, and the average length is 40 seconds. We use the official dataset split and report the result on the test-public set.

**Implementation Details.** We fine-tune the pre-training model from stage two on downstream video question-answering tasks following [23]. We uniformly sample 32 frames from each video. For How2QA [9], we concatenate the question with each answer from candidates, as well as subtitles. On top of the global [CLS] token of the question, we train an MLP to predict the confidence of each answer to be correct with a cross-entropy loss. For ActivityNet QA [24], we encode the answers in a one-hot fashion and train an MLP classifier on top of the global [CLS] token of the question overall answer candidates with a cross-entropy loss. For VIOLIN [11], we concatenate the hypothesis and subtitles, and on top of the global [CLS] token, we predict the confidence of the premise entails the hypothesis with a binary cross-entropy loss. We fine-tune the model with 8 NVIDIA Tesla V100 GPUs.

## C.3    Long-form Video Classification

**Datasets.** We conduct video classification tasks on two datasets which are consist of long-form videos. **(1) COIN** [19] is a large-scale dataset for comprehensive instructional video analysis. It consists of 11,827 videos related to 180 different tasks in 12 domains (e.g., vehicles, gadgets, etc.) related to our daily life. We use the official split and report the result on the test set. **(2) LVU** [21] is a benchmark that contains 9 tasks for evaluating long-form video understanding. It contains ~30K videos from ~3K movies from MovieClips [5]. We choose content understanding (relationship, speaking style, scene/place) tasks for evaluation. We use the official split and report the result on test set.

**Implementation Details.** We follow the setting of previous video-language pre-training models [8, 14, 18, 22] used for video classification tasks. We only use the video encoder with a linear layer on the top and fine-tune the model for video classification with a cross-entropy loss. We fine-tune the model with 8 NVIDIA Tesla V100 GPUs.

---

[4]We try our best to collect videos from YouTube, however, some videos are no longer available. We have downloaded 1628 training videos, 341 validation videos, and 346 testing videos.

[5]We download the videos from `https://www.movieclips.com`. ~90% videos can be downloaded while others are broken.

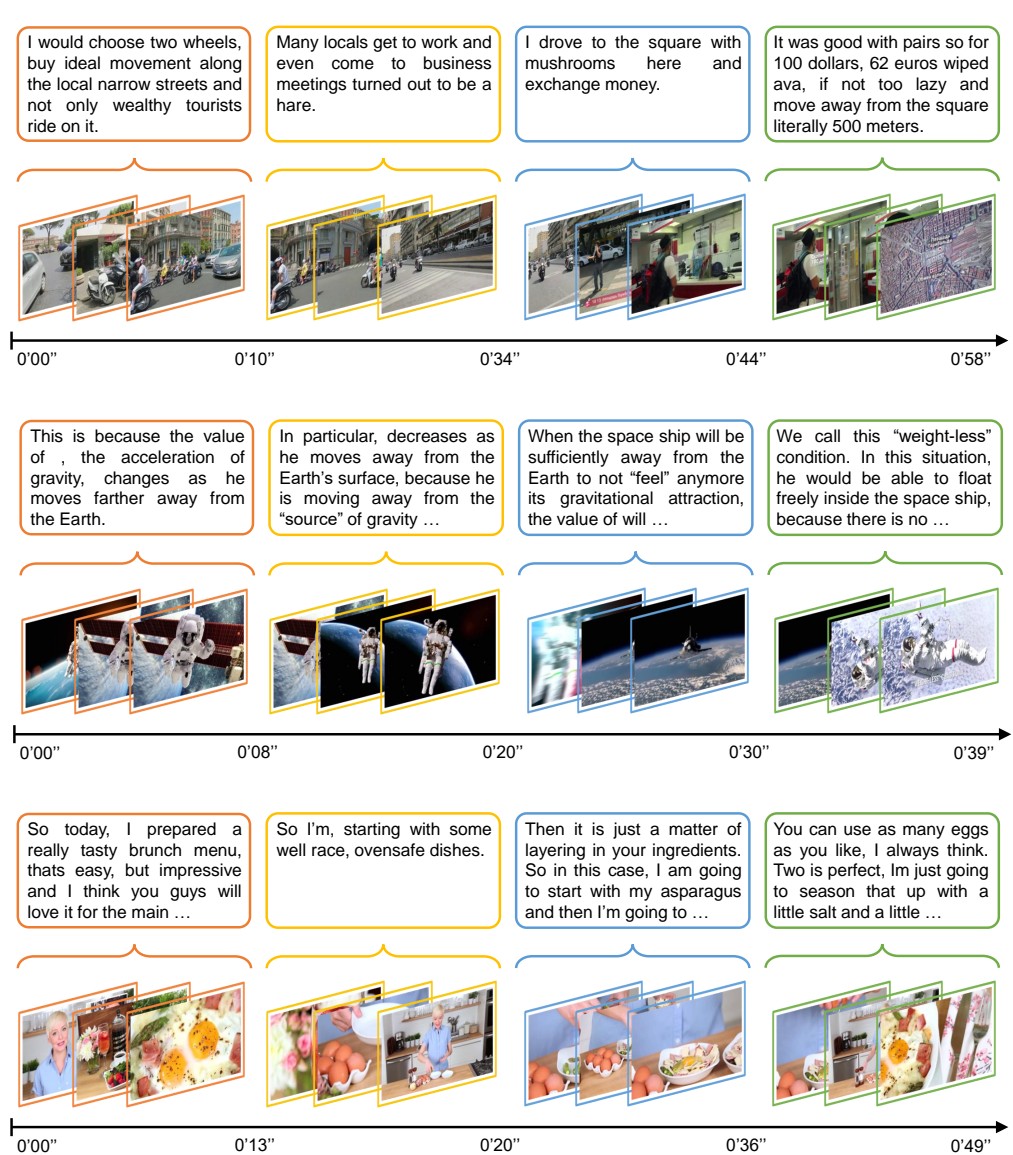

Figure 1: More examples of LF-VILA-8M.

# D  LF-VILA-8M Dataset Details

LF-VILA-8M is the largest long-form video-language dataset. We show more examples of long-form video-paragraph pairs in Fig. 1. In Fig. 2, we plot the histogram of video duration, text length, and the number of clip-sentence pairs.

# E  Limitation and Broader Impact.

This paper has a broader impact on many video-language understanding applications such as video-text retrieval, video question-answering, etc. Since we apply two-stream architecture in the first stage, we can also utilize the single-modality features (i.e., video and language) from our model for even broader tasks. By learning vision-language representation from unlabeled videos and subtitles,

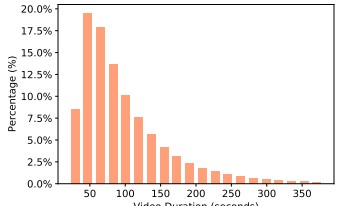 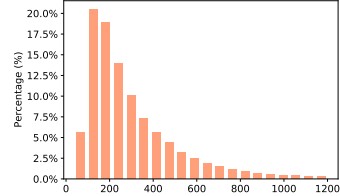 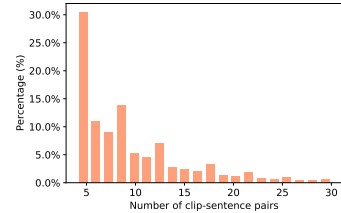

(a) Distribution of video duration.    (b) Distribution of text length.    (c) Distribution of the number of clip-sentence pairs.

Figure 2: More detailed statistics of LF-VILA-8M dataset.

our work may be easily extended and scaled to larger data. On the other hand, vision-language pre-training may learn biased or offensive content from user-generated video-subtitle data. This may cause an improper understanding of videos. However, these concerns are general to the entire field and are not amplified by this work.