# OpenReview forum: "Long-Form Video-Language Pre-Training with Multimodal Temporal Contrastive Learning"
_NeurIPS.cc/2022/Conference — NeurIPS 2022 Accept_

### Official Review · Reviewer_oRe6 · 2022-07-10

**Rating:** 7
**Confidence:** 3
**Soundness:** 3 good
**Presentation:** 3 good
**Contribution:** 3 good

**Summary:**

This paper proposes a method for long-form video-language pre-training by a multimodal temporal contrastive learning that leverages temporal continuity of clips within videos and their corresponding textual descriptions, and a hierarchical temporal window attention for encoding long videos using transformers. In addition, this paper proposes a new dataset with long-form video and paragraphs useful for pre-training video-language models. In experiments, the authors show that their method outperforms baselines in seven downstream tasks.

**Questions:**

Please see weaknesses.

**Limitations:**

Yes

**Strengths And Weaknesses:**

Strengths:
+ New dataset for long-form video-language modeling that will be released for research
+ New neural network architecture and contrastive pre-training for long-form video-language models
+ Outperforms baselines.


Weaknesses:

- Baselines not pre-trained with proposed dataset
-- In Table 1, the authors compare the proposed method trained with the proposed dataset outperforming the baselines on ActivityNet Captions. However, the baselines are not pre-trained with the same dataset as the proposed method. This makes the result inconclusive as to whether the performance boost comes from the proposed LF-VILA-8M dataset or the LF-VLP method or both. I suggest the authors to add results from the baselines pre-trained using LF-VILA-8M so that readers can make a stronger conclusion about the proposed method. Also, are all other results using baselines not pre-trained using LF-VILA-8M?


- Importance of HTWA.
-- In Table 5.b, the authors show a comparison between not using HTWA and using fixed size windows of 4 and 32. The difference between fixed 4 and HTWA performance seems marginal. Can the authors clarify how this result validates the effect of HTWA?

- Without contrastive loss on global representation?
-- In Table 5.a, the authors clearly show that pre-training is important. They also show the difference between using the proposed contrastive learning that uses time continuity to select the negatives and positives. However, I think this table is missing an experiment where the global contrastive learning is not used. This would make it clear if global contrastive learning is an important piece of the equation or if the proposed contrastive learning is good enough on its own.

- Use stage two for retrieval to compare the performance of the cross modal encoder?
-- The authors mention the representations learned from the model in stage one are widely used and efficient for retrieval. However, it would be good to report whether the cross modal encoder performs as well or better than the stage one model, especially since the cls token of this encoder outputs a score denoting whether the input paragraph and video are matched.

---

> ### Author Response · Authors · 2022-08-02
> **Response To Reviewer oRe6**
>
> We sincerely thank your constructive comments. We address your major concerns as below.
>
> **1. Baselines not pre-trained with proposed dataset.**
>
> LF-VILA-8M is a new dataset proposed by this paper, which is different from previous video-language dataset in consideration of video and language length. Previous models are designed for short-form video-language understanding while having difficulty to train on long-form videos (as shown in the following table).
>
> |Model|#Frames|GFlops|Mem|
> |:-:|:-:|:-:|:-:|
> |Frozen [3]|8|356|4.8G|
> |HD-VILA [43]|8|516|6.3G|
> |Frozen [3]|32|1424|11.6G|
> |HD-VILA [43]|32|1750|13.1G|
> |LF-VLP(Ours)| 32| **298**|**5.2G**|
>
> Thus we show our contributions by series of ablation studies in the paper (Table 5). To better clarify the source of the performance gain, we further conduct ablation studies on sampled 1M video-paragraph pairs from LF-VILA-8M dataset. We first show that **using more frames is better for long-form video-language understanding**. Also **pre-training on long-form video-language data further improves the performance significantly, which is first proposed in this work.** Further more, our **HTWA design enables the video encoder to encode more frames** and the **pre-training on long-form video with our MTC loss** both contribute to our performance. Please refer to our comments to the first reviewer (**More ablations**) for detailed experiment results and analysis.
>
> **2. Importance of HTWA.**
>
> HTWA steadily improves performance on all R@1, 5, 10 with negligible computational increase. For more thoroughly evaluation, we further conduct experiments on QuerYD dataset which consists of longer videos (278s on average)  and the results are showed as follows. We also conduct Classification of Procedural Activities on the COIN dataset.
> These results demonstrate the superiority of HTWA on videos with longer duration. We will include this important ablation study in our final version.
>
> | Method | TW# |  | QuerYD [31]|  | COIN [R1]|
> |---|---|---|:---:|---|:---:|
> |  |  | R@1 | R@5 | R@10 |  |
> | Fixed | 4 | 49.1 | 71.1 | 78.6 | 78.9 |
> | HTWA | [2-32] | **52.3** | **76.0** | **84.1** | **82.0** |
>
>
> **3. Without contrastive loss on global representation..**
>
> As our framework of Figure 2(a) shows, the $L_{time}$ is calculated at clip- and sentence-level representation to enhance the fine-grained alignment between long-video and paragraph. In Table 5(a) of the paper, we show the effectiveness of this loss. With only $L_{time}$, the fusion part in the model's high layers is not optimized. By the reviewer's suggestion, we show the result of pre-training using only $L_{time}$. We can draw the same conclusion from the table below.
>
> | Loss | R@1 | R@5 | R@50 |
> |---|---|---|---|
> | wo / Pre-training | 15.0 | 40.2 | 85.8 |
> | $L_{time}$ | 16.5 | 41.8 | 86.6 |
> | $L_{global}$ | 26.1 | 56.7 | 92.7 |
> | $L_{global}+L_{time}$ | **27.8** | **58.3** | **92.8** |
>
> **4. Use stage two for retrieval to compare the performance of the cross modal encoder.**
>
> We report retrieval results with model pre-trained in stage one due to two reasons. First, VTM score in stage two is based on a multimodal Transformer instead of dot-production in stage one, leading the computational complexity of retrieval to O(N**2). This makes it much less valuable for applications. Second, most existing works (e.g. Frozen, HD-VILA, Support Set) just use the dot-production of features. To make fair comparisons, we only do retrieval on stage one.
>
> As the reviewer's suggestion, we report the result of using cross-modal encoder (stage2 of LF-VLP) for retrieval. We finetune the stage2 model with contrastive loss and VTM loss. To speed up inference, we follow BLIP [R2] and first select 100 candidates based on the video-text similarity computed by stage1 model, and then using VTM score to rerank. The table below shows the result of ActivityNet dataset.
>
> | Method | R@1 | R@5 | R@50 |
> |---|---|---|---|
> | LF-VLP (Stage1) | 35.3 | 65.4 | 95.0 |
> | LF-VLP (Stage2) | 35.7 | 65.1 | 95.1 |
>
> The result shows the stage1 model performs as well as stage2 model. Consider the high efficiency of the stage1 model, it is the better choice for retrieval.
>
> > R1: COIN: A Large-scale Dataset for Comprehensive Instructional Video Analysis. CVPR, 2019.
>
> > R2: BLIP: Bootstrapping Language-Image Pre-training for Unified Vision-Language Understanding and Generation. ICML, 2022.

---

> > ### Comment · Reviewer_oRe6 · 2022-08-08
> > **Thank you**
> >
> > I would like to thank the authors for their efforts. Here are my comments.
> >
> > 1) While I understand that the dataset is a contribution of this work, it is not the main contribution and it is unfair to not train the other methods with the same dataset. This would be a topic we need to discuss with the other reviewers whether this is a factor or not in our final decision.
> >
> > 2) The authors have clarified my concerns about the need for HTWA. Thank you!
> >
> > 3) The authors have clarified my concerns about L_{time}. Thank you!
> >
> > 4) The authors have clarified my concerns about retrieval with stage 2 model. Thank you!
> >
> > I am increasing my score as the authors addressed almost all my concerns. There are still discussions needed between reviewers though so this may not be final.

---

> > > ### Author Response · Authors · 2022-08-09
> > > **Response To Additional Comments from Reviewer oRe6**
> > >
> > > Thank you very much for your recognition of this paper and this discussion. We will carefully add the discussion to the final version. Below is our further response to your concern:
> > >
> > > **Unfair to not train the other methods with the same dataset.**
> > >
> > > We conduct ablation studies on sampled 1M video-paragraph pairs from LF-VILA-8M dataset to verify the effectiveness of our method compared with previous baseline methods.
> > > To make a fair comparison, we keep most setting the same as other baseline methods (e.g., HD-VILA [43] and Frozen [3]) except for specifical designs for their pre-training dataset (e.g., we remove the high-resolution branch of HD-VILA), and we use the same backbone model as LF-VLP.
> > > We sample 8 frames from 4 continuous clips as the previous methods use about 8 frames for training. This is essentially the previous methods but use long-form video data for pre-training. Exp2 of the below table shows the result.
> > >
> > > |ID|#Pre-training  clips|#Frames in Total||ActivityNet [19]|||QuerYD [31]||
> > > |:-:|:-:|:-:|:-:|:-:|:-:|:-:|:-:|:-:|
> > > ||||R@1|R@5|R@50|R@1|R@5|R@10|
> > > |1|1|8|19.3|45.5|87.1|37.4|62.3|71.3|
> > > |2|4|8|20.3|47.0|87.7|42.3|66.2|74.6|
> > > |3|1|32|24.3|53.5|92.1|45.7|71.9|80.9|
> > > |4|4|32|**27.8**|**58.3**|**92.8**|**52.3**|**76.0**|**84.1**|
> > >
> > > Exp1, Exp3, and Exp4 show the effectiveness of our HTWA design for enabling the video encoder to encode more frames and the pre-training on long-form video with our MTC loss, please refer to our comments to the first reviewer (More ablations) for detailed experiment results and analysis.
> > >
> > > Additionally, we add Exp2 for fair comparison with previous methods. By comparing Exp2 and Exp4, it can be seen that the previous methods of using sparsely sampled frames and global contrastive loss to align videos and paragraphs have poor performance for long video retrieval. This shows that pre-training on long-form video not only needs long-form video data but also special design of methods.

---

### Official Review · Reviewer_1qEi · 2022-07-10

**Rating:** 6
**Confidence:** 3
**Soundness:** 3 good
**Presentation:** 3 good
**Contribution:** 3 good

**Summary:**

This paper focuses on pre-training a long-form video-language deep learning model. To handle the challenges in modeling long-form data, the authors propose 1) a large-scale dataset, 2) temporal contrastive loss, and 3) a hierarchical attention strategy. The proposed approach achieves favorable results on the retrieval and VQ tasks.

**Questions:**

Please see Weakness

**Limitations:**

Ths authors addressed the limitations and potential negative social impact well.

**Strengths And Weaknesses:**

*Strengths*
1. Except for some indexing issues, the paper is well-written and easy to follow.
2. The authors build a large-scale long-form video-paragraph dataset and promise to release it to the community. As the field is not well-explored, the dataset could be a critical factor in stimulating further research in this field.
3. The proposed pre-training algorithm improves the performance on several downstream tasks.
*Weakness*
1. The indexing $v_{i, k}$ and $t_{i, k}$ in Section 3.2 are a bit confusing. The authors could consider to change the index to $v^i_k$ and $t^i_k$.
2. It is unclear how the authors sample the set $\mathcal{V}$ for the positive instance $t_{i, q^+}$. Specifically, if the set $\mathcal{V}$ only contains frames that are all far away from the anchor along the time axis, then the instance with index $q^+$ is also far away from the anchor. How does the proposed method handle this situation?
3. Recently, there have been some papers showing the benefit of leveraging a pre-trained text encoder for visual-language tasks, e.g., Imagen. Can the authors discuss/comment on the potential results of leveraging a well-pre-trained text encoder on tasks this paper focuses on?
Saharia el at., "Photorealistic Text-to-Image Diffusion Models with Deep Language Understanding."

---

> ### Author Response · Authors · 2022-08-02
> **Response To Reviewer 1qEi**
>
> We sincerely thank your constructive comments. We address your major concerns as below.
>
> **1. Indexing Issue.**
>
> Thanks for your suggestion. We will change the index as your suggestion to make the equation more clearly.
>
> **2. Sampling strategy of V for the positive instance.**
>
> We thank your thorough consideration. Here V is randomly sampled for the positive instance. Since we consider 4 clips (last ~1 minutes) for pre-training in this work, the positive instance and V are not far away from each other. However, when we extend this loss for even longer videos (e.g., 10 minutes), we will need to restrict the maximum distance between positive pairs. We will include this condition in the paper.
>
> **3. Leveraging pre-trained text encoder.**
>
> According to R1, leveraging a stronger text encoder often leads to better performance. We can also expect better performance by using it. While in this paper, to make fair comparison with existing works, we initialize our text encoder with a pre-trained BERT due to its wide adoption in the field of research.
>
> > R1: Dou, Zi-Yi, et al. An empirical study of training end-to-end vision-and-language transformers. CVPR 2022.

---

> > ### Comment · Reviewer_1qEi · 2022-08-08
> > **Feedback to author responses**
> >
> > The author responses address my questions raised in the review. Thank you for the nice and hard work.

---

> > > ### Author Response · Authors · 2022-08-09
> > > **Response to Reviewer 1qEi**
> > >
> > > Thank you very much for your recognition of this paper and this discussion. We will carefully add the discussion to the final version.

---

### Official Review · Reviewer_ZX7j · 2022-07-11

**Rating:** 4
**Confidence:** 5
**Soundness:** 3 good
**Presentation:** 2 fair
**Contribution:** 2 fair

**Summary:**

This paper proposes a method for Long-Form Video-Language Pre-Training, in which the author introduce 1) multimodal temporal contrastive loss for aligning multiple modalities and 2) hierarchical temporal window attention for capturing spatiotemporal dependencies at lower cost. Experimental results demonstrate improvement over the previous state-of-the-art.

**Questions:**

1.	The multi-model in this paper refer to the vision and language. Is audio data useful or complementary in the proposed method?

**Limitations:**

1.	The large-scaled self-supervised pre-training may generate more carbon emission.

**Strengths And Weaknesses:**

Strengths:

1.	This paper is well written, and the core idea is easy to understand. The proposed method and formulate is clean, straightforward, and easy to re-implement.

2.	Long-term video modelling and multi-model alignment are unsolved problems in the community. They are challenging due to the video redundancy/complexity and memory burden. It is great to see this work trying to investigate these problems.

3.	The proposed hierarchical temporal window attention is an interesting direction for reducing the computation cost of the video transformer, which build local temporal attention window in early stage and global temporal dependencies in later stages when the feature map is smaller.

4.	The experimental result looks good, which outperform previous state-of-the-art works.

Weaknesses:

1.	In multimodal temporal contrastive loss, is there an assumption that the clip content and sentence will not repeat in the video? Otherwise, the line 50-51 will not make sense.

2.	Line 157-158, what is the sampling strategy here for K and V. If these two sets are far away from each other, samples in the positive pair may not match each other.

3.	The effectiveness of each component is not well analysed. There are some ablation studies in the Table 5. However, it is not clear to see the effectiveness of L_time alone, L_mlm, L_vtm, \lambda_1 and \lambda_2. It will be great to show how each part contribute to the final performance.

4.	Missing details in the comparison: Table 1-3 demonstrate the performance of this paper and previous works. In addition to the pretraining dataset, these details are also important: a). Computational cost. 2) Running Time. 3) Additional dataset (used in pretraining and pretrained model). 4) Input resolution. Combined with the weakness No.3, it is hard to distinguish which part of this paper is the key for the superior performance.

---

> ### Author Response · Authors · 2022-08-02
> **Response To Reviewer ZX7j Part 2.**
>
> **5. Use of audio data.**
>
> R3 shows that adding audio features greatly improves performance. However, it is not the main focus in this paper. To make fair comparison with most existing video-language pre-training works, we focus on video-language learning in this paper.
>
>
> **6. Generating more carbon emission.**
>
> This a common issue for large-scale pre-training models. In this work, we have tried to design efficient backbone to reduce computational cost compared with exisiting works (e.g., HD-VILA).
>
> > R1: Conceptual Captions: A Cleaned, Hypernymed, Image Alt-text Dataset For Automatic Image Captioning. ACL, 2018.
>
> > R2: Microsoft coco captions: Data collection and evaluation
> server. arXiv, 2015.
>
> > R3: Multi-modal transformer for video retrieval. ECCV, 2020.

---

> ### Author Response · Authors · 2022-08-02
> **Response To Reviewer ZX7j Part 1.**
>
> We sincerely thank your constructive comments. We address your major concerns as below.
>
> **1. Assumption of MTC loss.**
>
> Yes, we assume that most video contents are in sequential and videos with static content are outliers with a very low proportion in the dataset. We will add this assumption clearly in the paper.
>
> **2. Sampling strategy for K and V.**
>
> We thank your thorough consideration. Here K and V are randomly sampled. Since we consider 4 clips (last ~1 minutes) for pre-training in this work, K and V are not far away from each other. However, when we extend this loss for even longer videos (e.g., 10 minutes), we will need to restrict the maximum distance between positive pairs. We will include this condition in the paper.
>
> **3. The effectiveness of each component.**
>
> **The effectiveness of $L_{time}$.**
> As our framework of Figure 2(a) shows, the $L_{time}$ is calculated at clip- and sentence-level representation to enhance the fine-grained alignment between long-video and paragraph. In Table 5(a) of the paper, we show the effectiveness of this loss. With only $L_{time}$, the fusion part in the model's high layers is not optimized. By the reviewer's suggestion, we show the result of pre-training using only $L_{time}$. We can draw the same conclusion from the table below.
> | Loss | R@1 | R@5 | R@50 |
> |---|---|---|---|
> | wo / Pre-training | 15.0 | 40.2 | 85.8 |
> | $L_{time}$ | 16.5 | 41.8 | 86.6 |
> | $L_{global}$ | 26.1 | 56.7 | 92.7 |
> | $L_{global}+L_{time}$ | **27.8** | **58.3** | **92.8** |
>
>
> **The effectiveness of $\lambda_1$.**
> $\lambda_1$ is the weight of $L_{time}$. We have tried several settings and found 1.0 is better. The results on ActivityNet Retrieval are showed as follows:
>
> | $\lambda_1$ | R@1 | R@5 | R@50 |
> |---|---|---|---|
> | 0.0 | 26.1 | 56.7 | 92.7 |
> | 0.1 | 27.4 | 56.8 | 92.1 |
> | 0.5 | 27.4 | 57.6 | 92.0 |
> | 1.0 | **27.8** | **58.3** | 92.8 |
> | 2.0 | 27.5 | 57.9 | 92.6 |
> | 10.0 | 26.3 | 56.8 | **93.2** |
>
> **The effectiveness of $L_{mlm}$, $L_{vtm}$ and $\lambda_2$.**
> These two loss are used for the second stage of pre-training. Since we have freezed text encoder and video encoder for stage 2, they do not affect video-language retrieval performance. We conduct ablations on ActivityNet QA task to verify the effectiveness of these two losses. We report the result of pre-training for 3 epochs to save time. $\lambda_2$ is the weight of $L_{vtm}$. We empirically set it to 10.0 to balance the loss by referring to previous vision-language pre-training works. We adopt them as they are widely used in previous vision-language pre-training works [9, 22] and we do not claim them as our contribution.
>
> | Loss | Acc |
> |---|---|
> | w/o stage2 pre-training | 38.0 |
> | $L_{mlm}$ | 38.8 |
> | $L_{mlm} + L_{vtm}$ | **39.5** |
>
>
> **4. Missing details.**
>
> Thank you for your suggestion, we list the additional details here and will add them to the paper in the final version. We compare our model with other large-scale end-to-end pre-training models.
>
> | Model | Pre-training dataset | #Training examples | Pre-training Cost | #Param | input resolution |
> |---|---|---|---|---|---|
> | HD-VILA [3] | HD-VILA-100M [3] | 100M | 65K GPU Hours | 310M | 640x1024 (1 frame), 160x256 |
> | Frozen [43] | CC3M [R1], WebVid2.5M [43], COCO [R2] | 6.1M | 1.3K GPU Hours | 223M(181M*) | 224x224 |
> | LF-VLP (Ours) | LF-VILA-8M | 8M | 2.1K GPU Hours | 277M | 192x320 |
>
> \* means using distilled text encoder.
>
> Compared to Frozen which is pre-trained on human-annotated dataset, we achieve much better performance using relatively noisy but easy-to-get data. Compared to HD-VILA, we greatly reduce the training cost with a smaller model size. LF-VILA-8M is a subset of HD-VILA-100M (~60\% clips). Thus we achieve better performance with less pre-training data, fewer parameters and lower input resolution.
>
>
> To better clarify the source of the performance gain, we further conduct ablation studies on sampled 1M video-paragraph pairs from LF-VILA-8M dataset. We first show that **using more frames is better for long-form video-language understanding**. Also **pre-training on long-form video-language data further improves the performance significantly, which is first proposed in this work.** Further more, our **HTWA design enables the video encoder to encode more frames** and the **pre-training on long-form video with our MTC loss** both contribute to our performance. Please refer to our comments to the first reviewer (**More ablations**) for detailed experiment results and analysis.

---

### Official Review · Reviewer_rUgm · 2022-07-14

**Rating:** 4
**Confidence:** 4
**Soundness:** 3 good
**Presentation:** 3 good
**Contribution:** 2 fair

**Summary:**

In this paper, the authors proposed to pretrain an end-to-end vision-language model especially on long videos. A hierarchical temporal window based attention mechanism is proposed to deal with long temporal dependencies. The effectiveness of the proposed model is validated on extensive empirical evaluations.



**Questions:**

Please see the cons for details.

**Limitations:**

Yes

**Strengths And Weaknesses:**

Pros:
1. The idea of vision-language modeling on long-form videos is very interesting.
2. The paper is well presented.
3. Some strong empirical results are obtained.

Cons:
1. Important ablations are missing in the current evaluations.
a. The most comparable baseline HD-VILA is unfortunately not based on the same backbone model, which makes it hard to understand the source of the performance gain.
b. The authors entangled number of frames sampled and the temporal length in the current evaluation. What if only short video segments are used but the frames are densely sampled, e.g., using the HD-VILA-100M video segments but sampling 32 frames from each ~13-sec video segment.
2. Important existing benchmarks on long-form video understanding should have been also evaluated, including [41,5] and [r1, r2, r3]. These well-established benchmarks are important as they require even longer-term video inputs, which could be good test-bed to understand the generalization of the proposed model on longer videos, e.g., 3-4 minutes. More justification on the selection of downstream datasets should have been provided, including the length and complexity analysis of these datasets.


[r1] COIN: A Large-scale Dataset for Comprehensive Instructional Video Analysis, CVPR 2019
[r2] Learning To Recognize Procedural Activities with Distant Supervision, CVPR 22
[r3] Timeception for Complex Action Recognition, CVPR 19

---

> ### Author Response · Authors · 2022-08-02
> **Response To Reviewer rUgm Part 2.**
>
> **2. More benchmarks for evaluation.**
>
> **Justification on the selection of downstream datasets.**
> We highlight this work mainly for learning long-form video-language representation, thus we select important video-language tasks for evaluation. These datasets cover long-range videos, e.g., 180s for videos of ActivityNet, 278s for videos of QuerYD. We include details of these datasets in the supplementary material (Section C). We will add justification on the selection of downstream datasets in the main paper for final version.
>
> **Other long-form video understanding tasks.**
> By the reviewer's suggestion, we evaluate our video representation on COIN [R1] and LVU [41] benchmark. We follow the setting of previous video-language pre-training models used for video classification tasks. We only use the video encoder with a linear layer on the top and fine-tune the model for video classification.
>
> |Model|Pre-training Dataset|#Training Samples|Domain|Acc|
> |-|-|-|-|-|
> |ClipBERT [20]|COCO [R2], Visual Genome [R3]|5.6M|Open (out-domain)|65.4|
> |MIL-NCE [R4]|HowTo100M [30]|100M|Instructional (in-domain)|70.2|
> |VideoCLIP [R5]|HowTo100M [30]|100M|Instructional (in-domain)|72.5|
> |SlowFast [13]|Kinetics [7]|370K|Action (in-domain)|71.6|
> |TimeSformer [5]|Kinetics [7]|370K|Action (in-domain)|83.5|
> |TimeSformer [5]|HowTo100M [30]|100M|Instructional (in-domain)|85.3|
> |TimeSformer [5]|HowTo100M [30], wikiHow [R6]|100M|Instructional (in-domain)|**88.9**|
> |LF-VLP (Ours)|LF-VILA-8M|8M|Open (out-domain)|85.7|
>
> Above table shows the result of Classification of Procedural Activities on the COIN dataset in which the videos are instructional videos with an average duration of 141s. Our model achieves strong performance on this task, although we use out-domain videos for pre-training and our computational cost is smaller than the SOTA (~2.1K vs. 7K GPU hours). For this task, we largely surpass other video-language pre-training models (e.g., ClipBERT, MIL-NCE and VideoCLIP). We also outperform TimeSformer which uses Kinetics or HowTo100M for pre-training.
>
> |Model|Relation|Way of Speaking|Scene|
> |---|---|---|---|
> |R101-SlowFast+NL [13, R7, R8]|52.4|35.8|54.7|
> |VideoBERT [R9]|52.8|37.9|54.9|
> |Object Transformer [41]|53.1|39.4|56.9|
> |LF-VLP (Ours)|**61.5**|**41.3**|**68.0**|
>
> Above table shows the result of video classification on the LVU dataset which includes one-to-three-minute long videos. We surpass the previous SOTA methods largely, especially on Scene and Relation classification. Especially, we outperform the video-language pre-training model VideoBERT largely.
> The strong performances on these two benchmarks show the generalization power of our pre-trained video encoder.
>
> > R1: COIN: A Large-scale Dataset for Comprehensive Instructional Video Analysis. CVPR, 2019.
>
> > R2: Microsoft coco captions: Data collection and evaluation server. arXiv, 2015.
>
> > R3: Visual genome: Connecting language and vision using crowdsourced dense image annotations. IJCV, 2017.
>
> > R4: End-to-End Learning of Visual Representations from Uncurated Instructional Videos. CVPR, 2020.
>
> > R5: VideoCLIP: Contrastive Pre-training for Zero-shot Video-Text Understanding. EMNLP, 2021.
>
> > R6: Learning To Recognize Procedural Activities with Distant Supervision. CVPR, 2022.
>
> > R7: Deep residual learning for image recognition. CVPR, 2016.
>
> > R8: Non-local neural networks. CVPR, 2018.
>
> > R9: VideoBERT: A Joint Model for Video and Language Representation Learning. ICCV, 2019.

---

> > ### Comment · Reviewer_rUgm · 2022-08-05
> > **Very interesting results, please include in the final version**
> >
> > Thanks for the new evaluations!
> > These two new evaluations broaden the tasks and domains of the evaluation. Although the results on COIN did not achieve SOTA, this is actually very insightful for the community to understand possible limitations of this dataset and models pretrained on it. Therefore, adding these two new evaluations will actually make the paper stronger.

---

> > > ### Author Response · Authors · 2022-08-06
> > > **Response To Additional Comments from Reviewer rUgm**
> > >
> > > Thanks for your reply. We will include these two evaluations in our final version.

---

> ### Author Response · Authors · 2022-08-02
> **Response To Reviewer rUgm Part 1.**
>
> We sincerely thank your constructive comments. We address your major concerns as below.
>
> **1. Missing ablations.**
>
> **Using different backbone with HD-VILA.**
> We did not adopt the same backbone as HD-VILA on the new dataset because its video encoder cannot be fed with so many frames in long-form videos of LF-VILA-8M (4X length on average) while using more frames is critical which is shown later. In the table below, we measure the computational cost of our video encoder compared with HD-VILA and Frozen for encoding 8 or 32 frames. We claim the design of backbone specific for long-form video-language understanding as our contribution of this paper. For fair comparison, we show the effectiveness of our proposed methods using the same backbone in Table 5 of the paper.
>
> |Model|#Frames|GFlops|Mem|
> |:-:|:-:|:-:|:-:|
> |Frozen [3]|8|356|4.8G|
> |HD-VILA [43]|8|516|6.3G|
> |Frozen [3]|32|1424|11.6G|
> |HD-VILA [43]|32|1750|13.1G|
> |LF-VLP(Ours)| 32| **298**|**5.2G**|
>
> To clarify the source of the performance gain, we further conduct the following ablation studies on sampled 1M video-paragraph pairs from LF-VILA-8M dataset.
>
> **Using more frames is better for long-form video-language understanding.**
> Using the same backbone, we compare our model with pre-training with one clip-sentence pair for each sample as previous models (e.g., HD-VILA, Frozen). As shown in the table below, when only using 8 frames, the performance is poor. After increasing the number of frames to 32, there is a large improvement (**\#2 compared with \#1**). This shows that an efficient backbone to support more frames is essential for long-form video representation.
>
> |ID|#Pre-training  clips|#Frames in Total||ActivityNet [19]|||QuerYD [31]||
> |:-:|:-:|:-:|:-:|:-:|:-:|:-:|:-:|:-:|
> ||||R@1|R@5|R@50|R@1|R@5|R@10|
> |1|1|8|19.3|45.5|87.1|37.4|62.3|71.3|
> |2|1|32|24.3|53.5|92.1|45.7|71.9|80.9|
> |3|4|32|**27.8**|**58.3**|**92.8**|**52.3**|**76.0**|**84.1**|
>
>
> **Pre-training on long-form video-language further improves the performance significantly.** When we use 32 frames sampled from 4 continuous clips for pre-training, there is also a significant performance gain as shown in the above table (**\#3 compared with \#2**). The gain is larger on QuerYD dataset which consists of longer videos than ActivityNet (278s vs 180s on average). This shows the importance of pre-training on long-form video for longer video-language understanding. We are the first to propose pre-training with long-form video-language dataset and our model with a specially designed backbone enables the pre-training process.
>
> our **HTWA design enables the video encoder to encode more frames** and the
> **pre-training on long-form video with our MTC loss** both contribute to our performance. We will include these important ablation studies in our final version thanks to your suggestion.

---

> > ### Comment · Reviewer_rUgm · 2022-08-05
> > **Supporting evidence with a missing part**
> >
> > Thanks for the new ablation!
> > 1. Clarification questions: Is the experiment 2 essentially what was suggested in the review, i.e.,  using the HD-VILA-100M video segments but sampling 32 frames from each ~13-sec video segment? Is experiment 3 corresponding to LF-VLP in Table 1 and 2? If so, it seems there are some discrepancies.
> > 2. Missing part: It will be interesting to see the results of using 4 clips and 8 frames per clip. This variant could help to understand which contributes more, number of frames or number of clips.

---

> > > ### Author Response · Authors · 2022-08-06
> > > **Response To Additional Comments from Reviewer rUgm**
> > >
> > > Thanks for your reply.
> > >
> > > **1. Clarification questions.**
> > >
> > > **Is the experiment 2 essentially what was suggested in the review, i.e., using the HD-VILA-100M video segments but sampling 32 frames from each ~13-sec video segment?.**
> > >
> > > Yes. A clip is a ~ 13-sec video segment.
> > >
> > > **Is experiment 3 corresponding to LF-VLP in Table 1 and 2?**
> > >
> > > No. Exp3 is corresponding to LF-VLP of Table1 and Table2 but pre-trained on a small dataset. Exp3 is corresponding to the result of Table 5. To save computational cost, we conduct ablation studies on sampled 1M video-paragraph pairs from LF-VILA-8M.
> > >
> > >
> > > **2. Missing part: It will be interesting to see the results of using 4 clips and 8 frames per clip.**
> > >
> > > \#Frames means frames in total, so Exp3 uses 4 clips and 8 frames per clip. To make the results clearer, we change the table as follows:
> > >
> > > |ID|#Pre-training  clips|#Frames in Total||ActivityNet [19]|||QuerYD [31]||
> > > |:-:|:-:|:-:|:-:|:-:|:-:|:-:|:-:|:-:|
> > > ||||R@1|R@5|R@50|R@1|R@5|R@10|
> > > |1|1|8|19.3|45.5|87.1|37.4|62.3|71.3|
> > > |2|1|32|24.3|53.5|92.1|45.7|71.9|80.9|
> > > |3|4|32|**27.8**|**58.3**|**92.8**|**52.3**|**76.0**|**84.1**|

---

### Meta-Review · Area_Chair_srB7 · 2022-08-24

**Recommendation:** Accept
**Confidence:** Certain

**Metareview:**

After a discussion, the reviewers reach a consensus towards the acceptance. The author rebuttal resolved most of the concerns that the reviewers raised and the reviewers find its value to the community. Together with some of the reviewers, I also appreciate that the paper explores long-term video modelling which is an under-explored field. In addition to the long-form video pretraining technique, the paper also introduces a pretraining dataset with long-form videos showing the pretrained model achieves competitive performances on multiple retrieval and VideoQA benchmarks.

**Award:**

No

---

### Decision · Program_Chairs · 2022-09-14

Accept